# Beyond Direct Relations: Exploring Multi-Order Label Pair Dependencies for Knowledge Distillation

Submission Id: 2272

## ABSTRACT

Multi-label image classification is crucial for a wide range of multimedia applications. To address the resource limitation issue, various knowledge distillation (KD) methods have been developed to transfer knowledge from a large network (referred to as the "teacher") to a small network (referred to as the "student"). However, existing KD methods do not explicitly distill the dependencies between labels, which limits the model ability to capture multi-label correlation. Furthermore, although existing methods for multi-label image classification have utilized the second-order label pair dependency (direct dependency between two labels), the high-order label pair dependency, which captures the indirect dependency between two labels, remains unexplored. In this paper, we propose a **M**ulti-Order Label Pair **D**ependencies **K**nowledge **D**istillation (MDKD) framework. MDKD explicitly distills the knowledge to capture multi-order dependencies between labels, including the label pair dependencies from second-order and high-order, thus transferring the insight of label correlations from different perspectives. Extensive experiments on Pascal VOC2007, MSCOCO2014, and NUS-WIDE demonstrate the superior performances of MDKD.

## CCS CONCEPTS

• **Computing methodologies → Object recognition**.

## KEYWORDS

Knowledge distillation; Multi-label image classification; Indirect dependency

## 1 INTRODUCTION

Natural scenes in the real world often cover multiple objects. For example, as shown in Figure 1, an image from a street scene may include annotations for labels such as *person, dog, car*, and *parking meter*. Multi-label image classification aims to recognize each of these labels concurrently, which is crucial for a wide range of multimedia applications, including recommendation systems [27], image retrieval [23], and scene understanding [21]. To capture complex label features, there is always a necessity to train large models to obtain desirable performance in multi-label image classification.

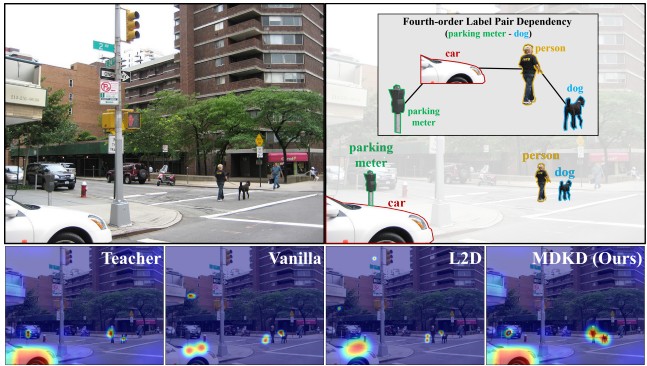

**Figure 1: An illustration of fourth-order label pair dependency formed by *parking meter-car-person-dog,* and the corresponding attention maps by different methods: 1) Vanilla: the backbone trained without distillation; 2) L2D: a logits-based method that achieves the state-of-the-art performance in KD for multi-label image classification; 3) Teacher: the pretrained backbone used for distillation; 4) Ours: the MDKD framework that explicitly transfers the knowledge to recognize multi-order label pair dependencies during distillation.**

Despite the remarkable achievements in multi-label image classification facilitated by deep neural networks, deploying these substantial models on resource-constrained devices, such as mobile phones, poses a significant challenge due to limited computational capacity and the necessity for swift inference times. To address this issue, knowledge distillation (KD) [10] have emerged as a powerful strategy, aiming to enhance the performance of small networks (referred to as "students") through the knowledge from large networks (known as "teachers"). Recently, KD methods have shown promising results to boost the capabilities of multi-label image classification [13, 22, 25, 26]. However, a common shortfall among these methods is their lack of explicit focus on modeling the multi-label dependencies, such as second-order label pair dependencies, which restricts their ability to fully understand the intricate relationships among multi-label.

The second-order label pair dependency, which considers the direct label co-occurrence, has played a crucial role in reasoning the occurrence of multi-label. For instance, MLGCN [3] proposed modeling second-order label pair dependencies, achieved through a learnable static correlation matrix. Additionally, Ye *et al.* [28] introduced a dynamic graph model to handle occasional label correlations in different images, while CADM [3] proposed incorporating the class-aware graph to explore label correlations. However, despite these advancements in second-order label pair dependencies, the exploration into the high-order label pair dependencies remains

unexplored. High-order label pair dependencies extend beyond direct co-occurrences to consider indirect relations. For example, in a street scene scenario like Figure 1, the association between *car* and *person* and the co-occurrence of *person* and *dog* imply an indirect dependency between *car* and *dog*. This scenario reflects a third-order label pair dependency, considering the indirect dependencies with *person* as the mediate label. Moreover, given that a parking meter is often associated with a car in urban street scenes, the above third-order label pair dependency could be further extended to a fourth-order one which implies the indirect relations between *parking meter* and *dog*, composing of the mediate second-order label pair dependencies of *parking meter-car*, *car-person*, and *person-dog*. Additionally, it is worth noting that the label sequences with the same endpoint labels but different mediate labels give rise to multiple high-order label pair dependencies. That is, a fourth-order label pair dependency formed by label sequence *parking meter-car-person-dog* does not consider the same thing as that in *parking meter-person-car-dog*, as the former considers the mediate second-order label pair dependencies of *parking meter-car*, *car-person*, and *person-dog*, while the latter considers the mediate second-order label pair dependencies of *parking meter-person*, *person-car*, and *car-dog*.

In this paper, we propose a **M**ulti-Order Label Pair **D**ependencies **K**nowledge **D**istillation (MDKD) framework for multi-label image classification, which explicitly distills the knowledge to discern label pair dependencies at multiple orders, including the second-order and high-order ones. Specifically, MDKD measures the multi-order label pair dependencies directly from the label predicted probabilities, which requires no extra architecture and training strategy within the general framework of multi-label image classification. Then, the measured label dependencies from teacher and student are aligned to encourage the knowledge transfer of multi-order label pair dependencies. As shown in Figure 1, by learning to capture the multi-order label pair dependencies, our MDKD correctly recognizes the *parking meter*, *car*, *person*, and *dog*, matching the attention from the teacher, while the vanilla and L2D fail to achieve this.

The main contributions are three-fold as follows:

• We first explore the indirect label pair dependencies, denoted by high-order label pair dependencies, in the field of multi-label image classification.

• We propose an MDKD framework for multi-label image classification that transfers knowledge to identify label pair dependencies at multiple orders, including direct (second-order) and indirect (high-order) ones.

• Extensive experiments on Pascal VOC2007 [8], MSCOCO2014 [12], and NUS-WIDE [4] demonstrate the capability of our method.

## 2 RELATED WORK

### 2.1 Multi-label Dependency

Multi-label image classification has recently yielded significant attention due to its importance in multimedia applications. Existing methods for multi-label image classification focus on modeling the label correlations, which provides prior knowledge for multi-label

classification: MLGCN [3] proposed modeling the label dependencies by a learnable static correlation matrix. Considering the occasional label co-occurrence, Ye *et al.* [28] proposed a dynamic graph model, capable of capturing the occasional label correlations at various images. Moreover, Zhu *et al.* [33] proposed refining the dynamic label co-occurrence by specific scenes. Beyond second-order label dependencies (between two labels), Wu *et al.* proposed an adaptive hypergraph for modeling these intricate many-to-one high-order interactions. However, these methods still can not explicitly model the indirect label dependencies. In this paper, we harness the multi-order label indirect correlation into KD. It is also worth noting that while the existing method [24] proposed utilizing the high-order dependency in multi-label learning, it simply considers how a single label will be impacted by a group of labels. In our MDKD, the combination of multi-order label pair dependencies provides a more granular perspective about how a single label impacts another via different orders of dependencies (direct and indirect) under a multi-label context.

### 2.2 Multi-label Knowledge Distillation

KD [10] aims to transfer knowledge from a large teacher network to a simpler and smaller student network. Existing KD methods can be primarily divided into two categories: logits-based methods and feature-based methods. Logits-based methods concentrate on designing distillation loss to transfer knowledge in the logits: For instance, the mutual learning strategy [32] encourages concurrent training of the teacher and student networks, enhancing the learning process through a cooperative framework. On the other hand, TAKD [16] proposed the "teacher assistant"—an intermediate-sized network that promotes the knowledge transfer between the more complex teacher and the simpler student networks. Feature-based methods focus on distilling knowledge from intermediate feature layers: For instance, FitNet [1] proposed minimizing the similarity between the feature maps of teacher and student. Attention transfer [30] focuses on attention maps rather than direct feature map comparisons. ReviewKD [2] proposed using multiple layers in teachers to supervise one layer in students.

While KD has been demonstrated to be a powerful strategy for transferring knowledge to students, the direct transplantation of typical KD to multi-label image classification presents a significant challenge. Specifically, logits-based methods often obtain the predicted probabilities based on the softmax function. However, softmax function is not suitable for multi-label, because the sum of predicted probabilities may not equal one in multi-label image classification. Feature-based methods proposed aligning the feature maps between teacher and student. This approach, while effective in harnessing contextual information, inadvertently biases the model towards the predominant objects. Therefore, it leads to the marginalization of less prominent objects within the scene.

Previous efforts have explored performing KD in multi-label image classification: Xu *et al.* [25] proposed a complementary parallel self-distillation to learn the joint patterns and the category-specific patterns of labels. Song *et al.* [22] proposed estimating the uncertainty of prediction to handle difficult labels. Liu *et al.* [13] proposed to enhance the label prediction by a weakly-supervised detection.

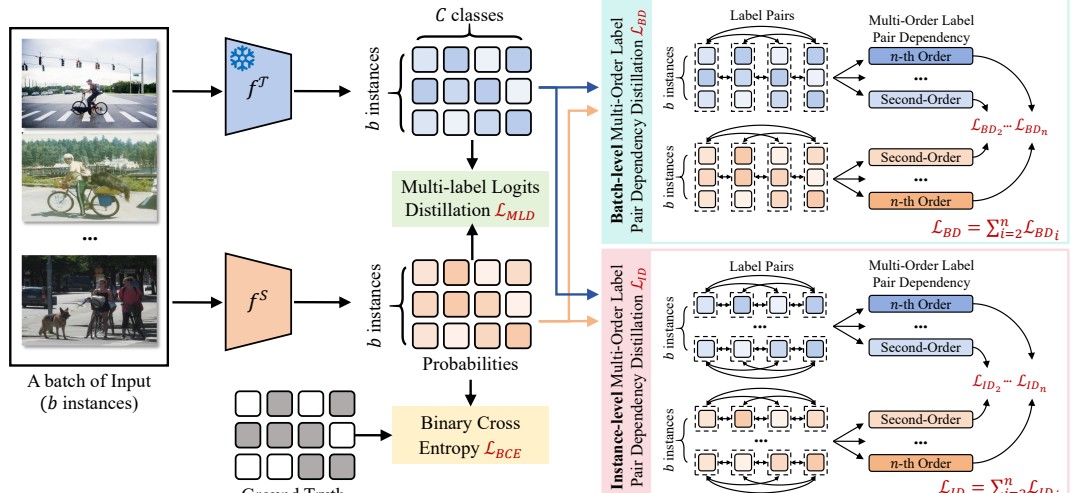

Figure 2: An overview of MDKD framework. The multi-order label pair dependencies are measured from the label predicted probabilities, where all possible label sequences are considered. The label dependencies from teacher and student are aligned for the knowledge transfer in recognition of label pair dependencies.

Yang *et al.* [26] proposed using the feature compactness in label-wise embeddings of the same classes, as well as the dispersion across different classes. Moreover, Yang *et al.* proposed a Multi-Label Logits Distillation (MLD) loss, which divides multi-label tasks into several binary classification problems. This stems from the understanding that the probabilistic of labels do not sum to one in a multi-label image. However, existing methods ignore explicitly modeling the distillation of label dependencies. To address this problem, we explore the distillation of label dependencies in this paper.

## 3 METHODOLOGY

Given a data sample $\{(\mathbf{x}, y)\}$, we have $\mathbf{x} \in \mathcal{X}$ with its corresponding label vector $y \in \mathcal{Y}$, where $\mathcal{X} \subset \mathbb{R}^d$ is the $d$-dimensional input space and $\mathcal{Y} \subset \{0, 1\}^C$ denotes the target space with $C$ classes. For each instance $\mathbf{x}$, the $j$-th component $y_j$ of $y$ is used to indicate whether $y_j$ is relevant to $\mathbf{x}$ or not. Specifically, $y_j$ is relevant to $\mathbf{x}$ when $y_j = 1$, and conversely $y_j$ is irrelevant to $\mathbf{x}$ if $y_j = 0$.

Figure 2 illustrates the MDKD framework. Given a batch of input instances (images), and the label indices $\{1, 2, \cdots, C\}$, the backbone network $f$ outputs a batch of predicted probabilities from teacher model $\{(\hat{y}_{i1}, \hat{y}_{i2}, \ldots, \hat{y}_{iC})\}_{i=1}^{b}$, where $C$ represents the total number of labels and $b$ denotes the batch size. $\hat{y}_{ik}$ denotes the predicted probability for $k$-th label within $i$-th instance. Note that the notation used in this paper takes superscripts $\mathcal{T}$ and $\mathcal{S}$ to denote the teacher and the student models. For example, $\hat{y}_{ik}^{\mathcal{T}}$ and $\hat{y}_{ik}^{\mathcal{S}}$ represent two label predicted probabilities computed from the teacher network $f^{\mathcal{T}}$ and the student network $f^{\mathcal{S}}$, respectively.

Following previous KD methods [10, 16, 26], MDKD integrates supervision loss between student output and the ground truth, and the soft label distillation loss, which encourages the knowledge transfer from teacher to student. For a batch of samples $\{(\mathbf{x}_i, y_i)\}_{i=1}^{b}$, the ground truth loss is implemented by Binary Cross Entropy (BCE) loss, encouraging the label predicted probabilities of the student network to be near to the ground truth:

$$\mathcal{L}_{\text{BCE}} = -\frac{1}{b} \left( \sum_{i=1}^{b} \sum_{k=1}^{C} y_{ik} \log \left( \hat{y}_{ik}^{\mathcal{S}} \right) + (1 - y_{ik}) \log \left( 1 - \hat{y}_{ik}^{\mathcal{S}} \right) \right). \quad (1)$$

The soft label loss is implemented by multi-label logits distillation loss $\mathcal{L}_{\text{MLD}}$, which considers the fact that in multi-label learning, the label predicted probabilities of all classes should sum to one. It minimizes the divergence between the binary predicted probabilities of teacher and student, which is computed as:

$$\mathcal{L}_{\text{MLD}} = \frac{1}{b} \sum_{i=1}^{b} \sum_{k=1}^{C} \mathcal{D} \left( \left[ \hat{y}_{ik}^{\mathcal{T}}, 1 - \hat{y}_{ik}^{\mathcal{T}} \right] \| \left[ \hat{y}_{ik}^{\mathcal{S}}, 1 - \hat{y}_{ik}^{\mathcal{S}} \right] \right), \quad (2)$$

where $[\cdot, \cdot]$ is the concatenation operation. $\mathcal{D}(\cdot \| \cdot)$ represents the KL divergence $\mathcal{D}(P \| Q) = \sum_{x \in \mathcal{X}} P(x) \log \left( \frac{P(x)}{Q(x)} \right)$.

However, with the BCE loss and multi-label logit distillation loss, the label dependencies still can not be explicitly distilled to the student. MDKD aims at distilling the multi-order label pair dependencies discerned by the teacher from input instances. Specifically, MDKD includes a batch-level multi-order label pair dependency distillation loss $\mathcal{L}_{\text{BD}}$ and an instance-level multi-order label pair dependency distillation loss $\mathcal{L}_{\text{ID}}$. The motivation behind incorporating these two levels of dependency distillation comes from the observation that some labels exhibit consistent co-occurrence across all scenarios, such as *tennis racket* and *ball*, whereas others predominantly co-occur in specific contexts, for instance, *person* and *car* in street scenes. While the batch-level multi-order label

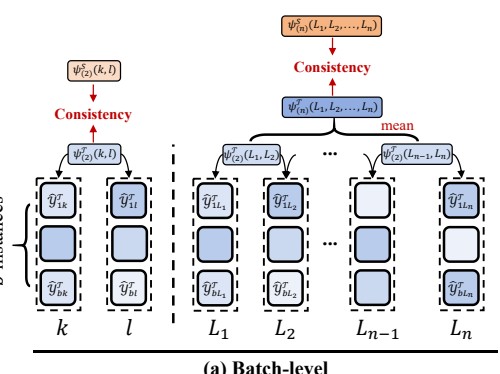 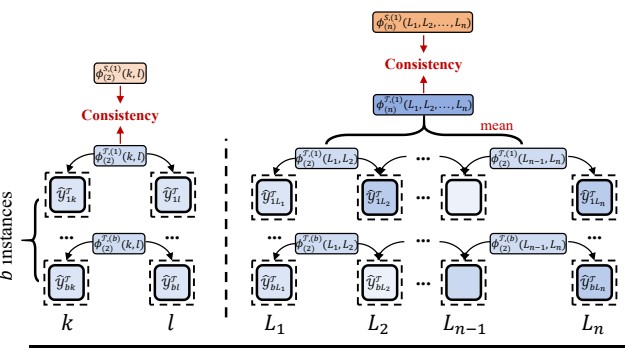

**Figure 3: An overview of batch-level and instance-level multi-order label pair dependency distillation.** $k, l, L_1, L_2, \cdots, L_n$ denote different label indices.

pair dependency distillation aims to encourage knowledge transfer of common multi-order label dependencies within a batch, the instance-level multi-order label pair dependency distillation transfers the knowledge in recognition of occasional multi-order label dependencies within an instance. These two levels of distillation loss collaboratively promote the knowledge transfer of label pair dependencies.

Overall, the total objective function of the proposed MDKD framework can be represented as:

$$\mathcal{L}_{\text{MDKD}} = \mathcal{L}_{\text{BCE}} + \lambda_{\text{MLD}}\mathcal{L}_{\text{MLD}} + \lambda_{\text{BD}}\mathcal{L}_{\text{BD}} + \lambda_{\text{ID}}\mathcal{L}_{\text{ID}}, \quad (3)$$

where $\lambda_{\text{MLD}}$, $\lambda_{\text{BD}}$, and $\lambda_{\text{ID}}$ are trade-off hyperparameters.

## 3.1 Batch-level Multi-Order Label Pair Dependency Distillation

Figure 3 (left) illustrates the process of batch-level multi-order label pair dependency distillation. Given the sets $\{\hat{y}_{1k}, \hat{y}_{2k}, \ldots, \hat{y}_{bk}\}$ and $\{\hat{y}_{1l}, \hat{y}_{2l}, \ldots, \hat{y}_{bl}\}$, where each represents the label predicted probabilities by the teacher model for $k$-th label and $l$-th label across various instances within a batch, respectively. The batch-level second-order label pair dependencies are measured through the mean square difference between two sets:

$$\Psi^{\mathcal{T}}_{(2)}(k, l) = \frac{1}{b} \sum_{i=1}^{b} (\hat{y}_{ik} - \hat{y}_{il})^2, \quad (4)$$

where $\hat{y}_{ik}$ and $\hat{y}_{il}$ denote the label predicted probabilities for $k$-th label and $l$-th label, respectively, for the $i$-th instance in a batch. Based on this measurement, the frequently co-occurring label pair exhibits similar label predicted probabilities across various instances, resulting in a lower mean square difference. We employ the Huber loss to measure the second-order label pair dependencies disparity between teacher and student, denoted as $\mho\left(\Psi^{\mathcal{T}}_{(2)}(k, l), \Psi^{\mathcal{S}}_{(2)}(k, l)\right)$, where $\mho(a, b)$ is defined as follows:

$$\mho(\mathbf{a}, \mathbf{b}) = \begin{cases} \frac{1}{2}(\mathbf{a} - \mathbf{b})^2 & |\mathbf{a} - \mathbf{b}| \leq 1, \\ |\mathbf{a} - \mathbf{b}| - \frac{1}{2} & \text{otherwise.} \end{cases} \quad (5)$$

It's worth noting that the dependencies $\Psi_{(2)}(k, l)$ and $\Psi_{(2)}(l, k)$ are considered the same for a lightweight complexity. This stems from the understanding that the label dependency is mutual, where the order slightly impacts the recognition of their dependency. Consequently, it can be easily proved that the total number of label pairs within second-order label pair dependency equals the combinations (i.e., half of the permutations). Suppose $P_{\mathbf{a}}^{\mathbf{b}} = \frac{\mathbf{a}!}{(\mathbf{a}-\mathbf{b})!}$ as the permutations, the batch-level second-order label pair dependency loss, $\mathcal{L}_{\text{BD}_2}$, which integrates the disparity of common second-order label pair dependencies within a batch between teacher and student, is defined as:

$$\mathcal{L}_{\text{BD}_2} = \frac{2}{P_C^2} \sum_{k,l \in [C], k \neq l} \mho\left(\Psi^{\mathcal{T}}_{(2)}(k, l), \Psi^{\mathcal{S}}_{(2)}(k, l)\right), \quad (6)$$

where $[C]$ represents the sequence of label indices $\{1, 2, \cdots, C\}$. $\frac{2}{P_C^2}$ denotes the reciprocal of the total number of second-order label pair dependencies, serving as a normalization for loss.

For the batch-level $n$-order label dependencies ($n \geq 3$), each batch-level $n$-order label pair dependency is measured by averaging the batch-level second-order label pair dependencies involving all intermediate label pairs. For example, given a fourth-order label sequence *parking meter-car-person-dog*, the batch-level fourth-order label pair dependency between *parking meter* and *dog*, articulated through this sequence, is derived by sequentially averaging the batch-level second-order label pair dependencies of *parking meter-car*, *car-person*, and *person-dog*. This stems from the understanding that a sequence of $n$-order label pair dependency is achieved through a series of mediate second-order label pair dependencies, creating a chain of dependencies that bridge the two endpoint labels. Given this, to present the formula for a sequence of batch-level $n$-order label pair dependency between the $L_1$-th and $L_n$-th labels, we define the label sequence as $\{L_1, L_2, \ldots, L_n\}$. Here, each element $L_k$ signifies a label index among the complete set of label indices

$\{1, 2, \ldots, C\}$. The batch-level $n$-th order label pair dependency, denoted by $\Psi_{(n)}(L_1, L_2, \ldots, L_n)$, is measured by:

$$\Psi_{(n)}(L_1, L_2, \ldots, L_n) = \frac{1}{n-1} \sum_{i=1}^{n-1} \Psi_{(2)}(L_i, L_{i+1}). \tag{7}$$

The batch-level $n$-order label pair dependency distillation considers the disparity of label pair dependencies captured by teacher and student for all unique label sequences, where the Huber loss $\mathcal{U}\left(\Psi_{(n)}^{\mathcal{T}}(L_1, L_2, \ldots, L_n), \Psi_{(n)}^{\mathcal{S}}(L_1, L_2, \ldots, L_n)\right)$ (as defined in Eq. 5) is employed to measure the disparity between teacher and student. Given $C$ labels, it is easy to prove that the total number of label pairs within the $n$-order is half of the permutation, i.e., $\frac{\mathrm{P}_C^n}{2}$. The batch-level $n$-order label pair dependency distillation loss, denoted by $\mathcal{L}_{\mathrm{BD}_n}$, is defined as:

$$\mathcal{L}_{\mathrm{BD}_n} = \frac{2}{\mathrm{P}_C^n} \sum_{j=1}^{\frac{\mathrm{P}_C^n}{2}} \mathcal{U}\left(\Psi_{(n)}^{\mathcal{T}}(L_1^j, L_2^j, \ldots, L_n^j), \Psi_{(n)}^{\mathcal{S}}(L_1^j, L_2^j, \ldots, L_n^j))\right), \tag{8}$$

where $\{L_1^j, L_2^j, \ldots, L_n^j\}$ denotes a $n$-order label sequence derived from the complete set of label indices $\{1, 2, \ldots, C\}$. Each element $L_k^j$ signifies a label index among the complete set of label indices $1, 2, \ldots, C$. $\frac{\mathrm{P}_C^n}{2}$ denotes the total number of $n$-order label pair dependencies, and its reciprocal serves as a normalization for loss. It is notable that when $n = 2$, this equation is with the same formulation as Eq. 6.

In the objective function of MDKD (Eq. 3), the part of batch-level multi-order label pair dependency loss is obtained by the weighted sum of global label dependency distillation loss from different orders:

$$\lambda_{\mathrm{BD}} \mathcal{L}_{\mathrm{BD}} = \sum_{i=2}^{n} \lambda_{\mathrm{BD}_i} \mathcal{L}_{\mathrm{BD}_i}, \tag{9}$$

where the series of $\lambda_{\mathrm{BD}_i}$ denotes the trade-off parameters.

## 3.2 Instance-level Multi-Order Label Pair Dependency Distillation

Different from the batch-level multi-order label pair dependency distillation, which transfers the knowledge to recognize multi-order label pair dependencies within a batch, the instance-level multi-order label pair dependency distillation loss considers the occasional label pair dependencies within different images. Figure 3 (right) illustrates the process of instance-level multi-order label pair dependency distillation. Given a batch of output probabilities from a teacher model, denoted as $\{(\hat{y}_{i1}, \hat{y}_{i2}, \ldots, \hat{y}_{iC})\}_{i=1}^{b}$. The instance-level second-order label pair dependency between $k$-th label and $l$-th label within $i$-th instance is measured by the square difference of their label predicted probabilities:

$$\Phi_{(2)}^{(i)}(k, l) \quad = \quad (\hat{y}_{ik} - \hat{y}_{il})^2. \tag{10}$$

This approach stems from the understanding that while the occasional label pair dependency is high, the square difference of label predicted probabilities within each instance is low.

The instance-level label dependency loss aggregates the difference of instance-level second-order label pair dependencies between teacher and student across the entire batch $b$, where the disparity of recognized instance-level second-order label pair dependencies are also measured by Huber loss (Eq. 5):

$$\mathcal{L}_{\mathrm{ID}_2} = \frac{1}{b} \frac{2}{\mathrm{P}_C^2} \sum_{i=1}^{b} \sum_{k,l \in [C], k \neq l} \mathcal{U}\left(\Phi_{(2)}^{\mathcal{T},(i)}(k, l), \Phi_{(2)}^{\mathcal{S},(i)}(k, l)\right). \tag{11}$$

For the instance-level $n$-order label dependencies ($n \geq 3$), each instance-level $n$-th order label pair dependency is measured by averaging the instance-level second-order label pair dependencies involving all intermediate label pairs. Consider a label sequence $\{L_1, L_2, \ldots, L_n\}$, which forms an $n$-order label pair dependency between the $L_1$-th and $L_n$-th labels. For the $i$-th instance, we define the instance-level $n$-order label dependency, denoted as $\Phi_{(n)}^{(i)}(k, l)$, in the following formula:

$$\Phi_{(n)}^{(i)}(L_1, L_2, \ldots, L_n) = \frac{1}{n-1} \sum_{k=1}^{n-1} \Phi_{(2)}^{(i)}(L_k, L_{k+1}). \tag{12}$$

The instance-level $n$-order label dependency distillation loss encourages the teacher model and student model to be consistent for the instance-level $n$-order label dependencies, formula as:

$$\mathcal{L}_{\mathrm{ID}_n} = \frac{1}{b} \frac{2}{\mathrm{P}_C^n} \sum_{i=1}^{b} \sum_{j=1}^{\frac{\mathrm{P}_C^n}{2}} \mathcal{U}\left(\Phi_{(n)}^{\mathcal{T},(i)}(L_1^j, L_2^j, \ldots, L_n^j), \Phi_{(n)}^{\mathcal{S},(i)}(L_1^j, L_2^j, \ldots, L_n^j)\right), \tag{13}$$

where $\sum_{i=1}^{b} \sum_{j=1}^{\frac{\mathrm{P}_C^n}{2}}$ ensures that all possible label sequences within each instance in a batch are considered in the loss function to measure and minimize the difference in dependency predictions between the teacher model and the student model. $\frac{\mathrm{P}_C^n}{2}$ is the total number of $n$-order label pair dependencies, and its reciprocal serves as the normalization for loss. It is notable that when $n = 2$, this equation is with the same formulation as Eq. 11. In the objective function of MDKD (Eq. 3), the part of instance-level multi-order label pair dependency distillation loss is obtained by the weighted sum of instance-level label dependency distillation loss from different orders:

$$\lambda_{\mathrm{ID}} \mathcal{L}_{\mathrm{ID}} = \sum_{i=2}^{n} \lambda_{\mathrm{ID}_i} \mathcal{L}_{\mathrm{ID}_i}, \tag{14}$$

where the series of $\lambda_{\mathrm{ID}_i}$ represents the trade-off parameters.

Given the detailed formulas in Eq. 9 and Eq. 14, the total objective function of MDKD in Eq. 3 can be re-written as:

$$\mathcal{L}_{\mathrm{MDKD}} = \mathcal{L}_{\mathrm{BCE}} + \lambda_{\mathrm{MLD}} \mathcal{L}_{\mathrm{MLD}} + \sum_{i=2}^{n} \lambda_{\mathrm{BD}_i} \mathcal{L}_{\mathrm{BD}_i} + \sum_{i=2}^{n} \lambda_{\mathrm{ID}_i} \mathcal{L}_{\mathrm{ID}_i}, \tag{15}$$

where the series of $\lambda_{\mathrm{ID}_i}$ and $\lambda_{\mathrm{BD}_i}$ are renewed trade-off hyperparameters for each specific label pair dependency distillation loss.

**Table 1: Performance on MS-COCO dataset for teacher and student models with the same network architectures.**

| Teacher | ResNet-101 | | | Swin-S | | | WRN-101 | | | WRN40-2 | | | ResNet56 | | |
|---|---|---|---|---|---|---|---|---|---|---|---|---|---|---|---|
| Student | ResNet-34 | | | Swin-T | | | WRN40-1 | | | WRN16-2 | | | ResNet20 | | |
| Metrics | mAP | OF1 | CF1 | mAP | OF1 | CF1 | mAP | OF1 | CF1 | mAP | OF1 | CF1 | mAP | OF1 | CF1 |
| Teacher | 73.02 | 73.81 | 68.24 | 82.18 | 79.86 | 77.44 | 74.96 | 75.68 | 71.01 | 75.42 | 77.11 | 74.68 | 72.18 | 74.26 | 68.09 |
| Student | 70.55 | 72.17 | 66.20 | 80.01 | 78.68 | 74.39 | 72.48 | 74.12 | 70.28 | 73.19 | 75.68 | 72.17 | 69.25 | 71.45 | 66.28 |
| RKD [17] | 70.68 | 72.32 | 66.34 | 80.45 | 79.11 | 74.82 | 73.98 | 75.25 | 71.42 | 73.18 | 75.62 | 72.15 | 69.48 | 71.98 | 67.18 |
| PKT [18] | 71.01 | 72.88 | 66.73 | 79.45 | 79.15 | 75.33 | 73.15 | 75.05 | 70.40 | 73.89 | 76.14 | 72.89 | 70.15 | 72.16 | 66.40 |
| ReviewKD [2] | 70.56 | 72.48 | 66.04 | 79.80 | 79.12 | 74.92 | 73.59 | 74.48 | 71.63 | 74.01 | 76.08 | 72.69 | 70.45 | 72.68 | 67.34 |
| MSE [25] | 70.80 | 72.54 | 66.15 | 79.48 | 80.02 | 76.15 | 72.15 | 74.05 | 71.88 | 74.22 | 76.15 | 73.05 | 69.40 | 72.44 | 67.48 |
| PS [22] | 70.18 | 72.00 | 67.89 | 80.33 | 80.94 | 76.09 | 72.65 | 74.33 | 72.19 | 73.04 | 75.65 | 73.05 | 69.77 | 71.18 | 66.34 |
| L2D [26] | 72.09 | 73.54 | 68.47 | 80.43 | 81.09 | 76.85 | 74.08 | 73.98 | 69.97 | 72.98 | 76.89 | 72.03 | 71.45 | 73.09 | 67.54 |
| **MDKD** | **73.35** | **74.13** | **68.89** | **82.33** | **82.17** | **77.56** | **76.88** | **77.45** | **74.18** | **75.34** | **77.20** | **73.89** | **73.37** | **74.08** | **68.29** |

**Table 2: Performance on MS-COCO dataset for teacher and student models with different network architectures.**

| Teacher | ResNet-101 | | | Swin-T | | | ResNet-101 | | | VGG13 | | | ResNet32×4 | | |
|---|---|---|---|---|---|---|---|---|---|---|---|---|---|---|---|
| Student | RepVGG-A0 | | | ResNet-34 | | | MobileNet v2 | | | MobileNet v2 | | | ShuffleNet-V2 | | |
| Metrics | mAP | OF1 | CF1 | mAP | OF1 | CF1 | mAP | OF1 | CF1 | mAP | OF1 | CF1 | mAP | OF1 | CF1 |
| Teacher | 73.02 | 73.81 | 68.24 | 80.01 | 78.68 | 74.39 | 73.02 | 73.81 | 68.24 | 74.08 | 75.05 | 72.20 | 79.44 | 79.04 | 75.98 |
| Student | 69.58 | 72.48 | 66.80 | 70.55 | 72.17 | 66.20 | 71.99 | 73.15 | 67.87 | 71.99 | 73.15 | 67.87 | 72.58 | 73.13 | 70.15 |
| RKD [17] | 70.55 | 72.98 | 67.05 | 71.13 | 72.68 | 66.89 | 71.56 | 73.18 | 67.99 | 72.48 | 73.48 | 67.48 | 72.00 | 73.27 | 70.49 |
| PKT [18] | 70.96 | 72.15 | 66.65 | 70.69 | 72.41 | 66.18 | 71.46 | 72.79 | 68.04 | 71.45 | 73.08 | 68.43 | 73.12 | 73.18 | 70.48 |
| ReviewKD [2] | 69.18 | 72.80 | 67.18 | 70.98 | 72.48 | 66.40 | 71.94 | 73.41 | 67.96 | 72.49 | 73.32 | 69.18 | 73.15 | 73.89 | 70.14 |
| MSE [25] | 70.18 | 73.05 | 67.45 | 71.08 | 72.84 | 67.01 | 71.59 | 73.18 | 68.04 | 72.00 | 73.90 | 68.87 | 72.14 | 73.64 | 71.00 |
| PS [22] | 70.69 | 72.44 | 67.39 | 71.32 | 72.05 | 66.84 | 71.15 | 73.04 | 67.54 | 71.43 | 72.98 | 68.78 | 72.45 | 73.81 | 70.78 |
| L2D [26] | 71.49 | 73.67 | 67.98 | 72.94 | 73.05 | 67.78 | 72.84 | 73.89 | 68.60 | 71.05 | 73.98 | 68.18 | 72.66 | 73.22 | 71.34 |
| **MDKD** | **72.65** | **74.66** | **68.42** | **74.01** | **74.99** | **68.49** | **73.40** | **74.45** | **69.41** | **73.98** | **74.89** | **71.80** | **74.07** | **75.13** | **72.66** |

**Table 3: Performance of the comparing methods on VOC dataset in terms of AP and mAP (%). The best results are marked in red, and the second-best results are marked in blue.**

| Methods | Vanilla | RKD[17] | PKT[18] | MSE[25] | L2D[26] | **MDKD** |
|---|---|---|---|---|---|---|
| bottle | 56.89 | 57.15 | 57.91 | 57.80 | 59.24 | 59.96 |
| pottedplant | 67.15 | 66.89 | 66.48 | 68.15 | 70.10 | 72.88 |
| chair | 70.45 | 71.15 | 71.07 | 70.90 | 73.59 | 75.14 |
| sofa | 73.26 | 74.15 | 73.88 | 73.48 | 73.48 | 75.89 |
| diningtable | 76.05 | 76.54 | 76.43 | 77.15 | 78.10 | 79.25 |
| cow | 82.11 | 82.15 | 81.89 | 82.45 | 82.98 | 84.44 |
| tvmonitor | 82.59 | 83.18 | 83.40 | 82.48 | 84.15 | 85.94 |
| bus | 85.15 | 84.48 | 85.00 | 86.14 | 85.66 | 87.15 |
| sheep | 83.09 | 83.15 | 83.45 | 84.27 | 84.98 | 85.18 |
| motorbike | 88.15 | 89.48 | 88.15 | 87.45 | 88.18 | 89.03 |
| dog | 90.97 | 89.11 | 91.64 | 90.59 | 91.97 | 93.01 |
| bird | 90.56 | 91.76 | 91.45 | 90.35 | 91.67 | 91.72 |
| bicycle | 91.55 | 91.15 | 90.48 | 92.15 | 91.98 | 92.45 |
| cat | 91.08 | 91.45 | 91.66 | 91.47 | 92.45 | 93.48 |
| boat | 92.12 | 91.45 | 92.57 | 92.66 | 93.02 | 92.99 |
| car | 92.45 | 92.10 | 92.77 | 92.89 | 93.40 | 93.73 |
| horse | 94.89 | 93.67 | 93.01 | 93.75 | 95.47 | 96.10 |
| person | 95.77 | 95.42 | 95.26 | 96.08 | 96.22 | 96.87 |
| train | 96.03 | 96.98 | 96.55 | 96.48 | 96.77 | 97.15 |
| aeroplane | 97.15 | 97.48 | 97.54 | 96.88 | 97.06 | 97.89 |
| mAP | 84.88 | 84.94 | 85.03 | 85.18 | 86.02 | 87.01 |

## 4 EXPERIMENT

### 4.1 Datasets

We perform experiments on MS-COCO2014 (MS-COCO for short) [12], Pascal VOC 2007 (VOC for short) [8], and NUS-WIDE [4], to validate the effectiveness of our proposed method. VOC includes 5,011 images in the train-val set, 4,952 images in the test set, and 20 distinct label categories with an average of 1.6 labels per image. NUS-WIDE contains 161,789 images and 107,859 test images, covering 81 label categories. MS-COCO contains 82,081 images for training and 40,137 for testing, with 80 label categories, an average of 2.9 labels per image.

### 4.2 Metrics

We follow L2D [26] to use the mean average precision (mAP) over all classes, average per-class F1-score (CF1), and overall F1-score (OF1) to evaluate the performance. The OF1 and CF1 consider both recall and precision.

### 4.3 Implementation Details

We utilize models pre-trained on the ImageNet dataset [5] as backbones. Images are uniformly resized to a resolution of 224x224 pixels. The batch size is set as 32. We adopt random horizontal flipping and Cutout [6] for augmentation. The Adam optimizer [11] is used to train the model for 110 epochs, starting with an initial learning rate of 0.00015 and applying cosine annealing to adjust the learning rate throughout the training process. The weight decay is set to 0.0001. The highest order of multi-order label dependency distillation is set as fourth-order. The $\lambda_{BD_2}$, $\lambda_{BD_3}$, $\lambda_{BD_4}$, $\lambda_{ID_2}$, $\lambda_{ID_3}$, and $\lambda_{ID_4}$ are all set as 1. Following L2D [26], we set the $\lambda_{MLD}$ as 10. For all feature-based methods, we perform distillation on the feature maps output from the visual backbone $f$. To validate the MDKD with various architectures of teacher and student models, we employ the common backbones including RepVGG [7], ResNet [9], Swin Transformer [14], Wide ResNet (WRN) [31], MobileNet v2 [19], and ShuffleNet-V2 [15].

**Table 4: Ablation studies for MLD, batch-level multi-order label pair dependency distillation and instance-level multi-order label pair dependency distillation on MS-COCO.**

| | MLD | Batch-level (-order) | | | | | Instance-level (-order) | | | | | Metrics | | |
|---|---|---|---|---|---|---|---|---|---|---|---|---|---|---|
| | | second- | third- | fourth- | fifth- | sixth- | second- | third- | fourth- | fifth- | sixth- | mAP | OF1 | CF1 |
| 1 | - | - | - | - | - | - | - | - | - | - | - | 70.55 | 72.17 | 66.20 |
| 2 | ✓ | - | - | - | - | - | - | - | - | - | - | 71.21 | 72.85 | 67.43 |
| 3 | ✓ | ✓ | - | - | - | - | ✓ | - | - | - | - | 71.85 | 73.28 | 67.66 |
| 4 | ✓ | - | ✓ | - | - | - | - | ✓ | - | - | - | 71.58 | 73.48 | 68.00 |
| 5 | ✓ | ✓ | ✓ | - | - | - | ✓ | ✓ | - | - | - | 72.87 | 74.01 | 68.15 |
| 6 | ✓ | - | ✓ | ✓ | - | - | - | ✓ | ✓ | - | - | 73.15 | 73.54 | 68.01 |
| 7 | ✓ | ✓ | ✓ | ✓ | - | - | - | - | - | - | - | 73.25 | 73.48 | 68.90 |
| 8 | ✓ | - | - | - | - | - | ✓ | ✓ | ✓ | - | - | 73.04 | 74.02 | 69.11 |
| 9 | ✓ | ✓ | ✓ | ✓ | - | - | ✓ | ✓ | ✓ | - | - | 74.15 | 75.05 | 70.12 |
| 10 | - | ✓ | ✓ | ✓ | - | - | ✓ | ✓ | ✓ | - | - | 72.69 | 73.00 | 68.19 |
| 11 | ✓ | ✓ | ✓ | ✓ | ✓ | - | ✓ | ✓ | ✓ | ✓ | - | 73.58 | 75.52 | 69.75 |
| 12 | ✓ | ✓ | ✓ | ✓ | ✓ | ✓ | ✓ | ✓ | ✓ | ✓ | ✓ | 74.26 | 74.79 | 69.85 |

## 4.4 Comparison Study

To evaluate our proposed method, we perform a comparison study with state-of-the-art KD methods, including the logits-based (RKD [17], MSE [25], PS [22], and L2D [26]) and feature-based (PKT [18] and ReviewKD [2]) methods. Table 1 and Table 2 demonstrate the comparison results on MS-COCO, evaluating the effectiveness of teacher-student model pairings with both the same and different architectures. Table 1 reveals that the MDKD method consistently outperforms others across three metrics. For instance, a notable improvement is observed when ResNet56 serves as the teacher and ResNet20 as the student, with MDKD achieving mAP improvements of 3.89% over RKD and 4.12% over the base student model. Moreover, MDKD frequently surpasses even the performance of teachers. This superior performance is evident across various backbone combinations, demonstrating the superior ability of MDKD. Additionally, Table 2 reveals that the disparity in performance between teacher and student models with differing architectures is larger than that when architectures are the same, yet MDKD still significantly outperforms other methods and occasionally exceeds the teacher model.

Furthermore, Table 3 shows the performance of different methods on VOC in terms of AP and mAP. It is shown that MDKD achieves the highest scores in most cases than other KD methods, including AP and mAP. Notably, for challenging classes such as *horse*, where other methods may underperform compared to the vanilla (without performing distillation), MDKD demonstrates a remarkable 1.21% AP improvement over vanilla. Table 6 (Appendix) presents comparison results on NUS-WIDE. It is shown that MDKD also outperforms other state-of-the-art methods in all cases, which demonstrates its superior ability. These experimental results highlight that while the state-of-the-art methods only transfer the knowledge within feature maps and logits, they achieve sub-optimal efficacy due to the lack of knowledge to capture label dependencies, while our method leverages label dependencies to enhance the KD in multi-label image classification tasks and achieves high performances.

## 4.5 Ablation Study

To further evaluate the ability of multi-order label pair dependency distillation, we conduct ablation studies on MS-COCO, employing

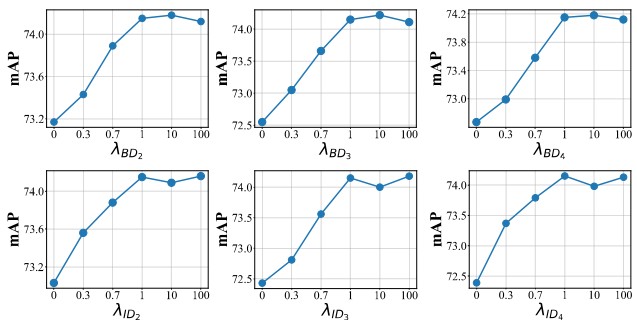

**Figure 4: Sensitivity analysis of $\lambda_{BD_2}$, $\lambda_{BD_3}$, $\lambda_{BD_4}$, $\lambda_{ID_2}$, $\lambda_{ID_3}$, and $\lambda_{ID_4}$.**

ResNet-101 as the teacher and ResNet-34 as the student. The results are shown in Table 4, where the first row presents the baseline performance of the student model without any distillation applied. The results in rows 2, 3, 5, and 9 demonstrate a step-wise enhancement with the implementation of second-order, third-order, and fourth-order label pair dependency distillation, before the implementation of fifth-order label pair dependency distillation. However, the distillation of fifth-order label pair dependencies and sixth-order label pair dependencies (as seen in the last two rows) does not significantly enhance the ability of the model. We argue that this phenomenon occurs because the correlation between label pairs becomes increasingly marginal when extended to overly high orders.

## 4.6 Sensitivity Analysis

This section performs a sensitivity analysis on the parameters $\lambda_{BD_2}$, $\lambda_{BD_3}$, $\lambda_{BD_4}$, $\lambda_{ID_2}$, $\lambda_{ID_3}$, and $\lambda_{ID_4}$ to further evaluate their impact on model effectiveness. As depicted in Figure 4, an increase in these parameters from 0 to 1 promotes an improvement in model performance. This observation reveals that the distillation of multi-order label pair dependencies can improve the model performance. Additionally, beyond the value of 1, the performance of the model is insensitive to their values.

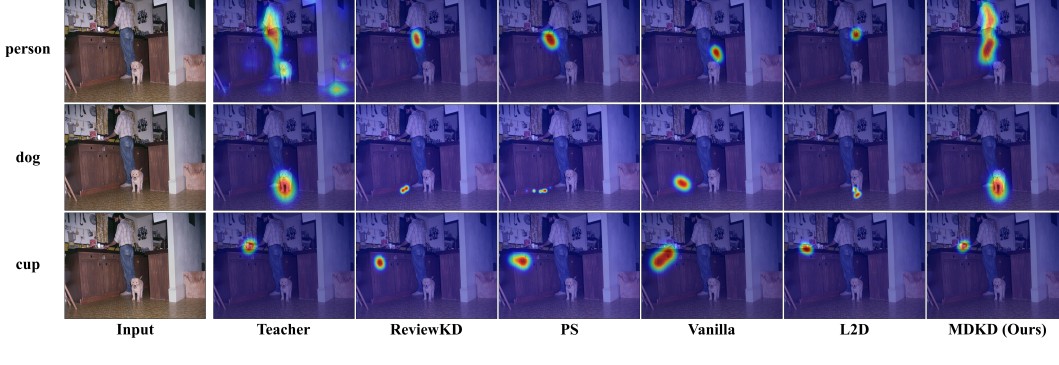

**Figure 5: Attention maps for *person, dog,* and *cup* by different methods.**

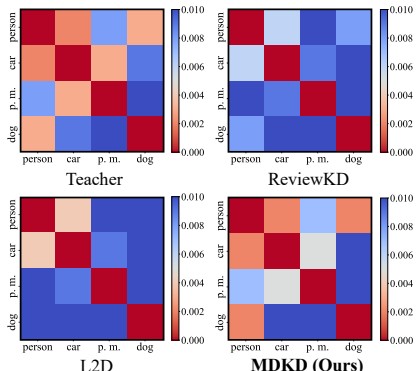

**Figure 6: The instance-level second-order label pair dependencies of different methods, measured by square difference (lower values indicating higher correlation). The scores among four labels: *parking meter* (p. m. for short), *car, person,* and *dog* are presented. The input image is shown in Figure 1.**

### 4.7 Visualization of Attention Map

To evaluate whether the student focuses on objects correctly corresponding to the teacher, we visualize the attention maps from different classes using Grad-CAM [20]. Our MDKD method demonstrates superior precision in recognizing objects compared to other methods, as illustrated in Figure 5 and Figure 7 (Appendix). For instance, in Figure 5, MDKD accurately captures the cup, while other methods exhibit recognition failures. The clear recognition of different classes reveals the superior ability of MDKD compared with other methods.

### 4.8 Label Dependencies Knowledge Transfer

To evaluate whether the knowledge to recognize multi-order label pair dependencies is better transferred in MDKD, we compare the mean square and square differences in output probabilities from different methods. The teacher is ResNet-101 and the student is ResNet-34. We demonstrate the second-order label pair dependencies in Figure 6 and Figure 8 (Appendix). The third-order label pair dependencies and fourth-order label pair dependencies are presented in Table 7 (Appendix) and Table 8 (Appendix). The results show that the label pair dependencies captured by MDKD

**Table 5: Performance of reversed distillation on the MS-COCO dataset.**

| Metrics | mAP | OF1 | CF1 |
|---|---|---|---|
| Teacher | 70.55 | 72.17 | 66.20 |
| Student | 73.02 | 73.81 | 68.24 |
| RKD [17] | 73.48 | 74.02 | 68.84 |
| PKT [18] | 73.92 | 73.48 | 68.15 |
| ReviewKD [2] | 73.84 | 74.11 | 69.41 |
| MSE [25] | 73.66 | 74.02 | 69.52 |
| PS [22] | 73.05 | 73.98 | 68.54 |
| L2D [26] | 74.52 | 74.67 | 69.18 |
| **MDKD** | **76.89** | **77.45** | **73.68** |

better match the ones from the teacher compared to other methods, which reveals its capability to transfer knowledge in recognition of multi-order label pair dependencies.

### 4.9 Reversed Knowledge Distillation

Previous study [29] has shown that even a high-performing model can be further improved through guidance from a less complex model. To delve deeper into the efficacy of MDKD within the context of reverse KD, we experiment by assigning ResNet-34 as the teacher and ResNet-101 as the student, resulting in the student surpassing the performance of the teacher. As shown in Table 5, MDKD still achieves higher scores than all other methods, underscoring its exceptional capability.

## 5 CONCLUSION

In this paper, we propose the MDKD framework, which explicitly transfers the knowledge to capture multi-order label pair dependencies during KD. The multi-order distillation transfers the insight to capture label correlations from different perspectives. Specifically, MDKD includes batch-level multi-order label pair dependency distillation, which considers the multi-order label pair dependencies within a batch, and instance-level multi-order label pair dependency distillation, which addresses occasional label multi-order label pair dependencies based on specific scenes. Extensive experiments on different datasets reveal the superior ability of MDKD.

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
