# OpenReview forum: "Beyond Direct Relationships: Exploring Multi-Order Label Pair Dependencies for Knowledge Distillation"
_acmmm.org/ACMMM/2024/Conference — MM2024 Poster_

### Official Review · Reviewer_JDdT · 2024-05-17

**Rating:** 3
**Confidence:** 3

**Summary:**

This paper tackles the multi-label classfication problem by exploreing multi-order label pair dependencies for knowledge distillation.

**Strengths:**

- The paper is clearly written and easy to follow.

**Limitations:**

- There is a general issue in the abstract. The authors claim that the high order label pair dependency remains unexplored. In many survey papers of multi-label leanring, they often divide existing methods into three categories according to the exploration of label correlations: first-order, second-order and high order. Thus, such description in abstact is too arbitrary.

- You should explain why your proposed method is superior to other high-order based method.

- In Figure 2, it is easy to understand the instance-level multi-order label pair, how to understand batch-level? Compared with instance-level, what are the superiorities of batch-level?

**Suitability:**

2

---

### Official Review · Reviewer_ufio · 2024-05-20

**Rating:** 3
**Confidence:** 3

**Summary:**

The paper proposes a Multi-Order Label Pair Dependencies Knowledge Distillation (MDKD) framework for multi-label image classification.The key idea is to explicitly distill both second-order (direct) and high-order (indirect) label pair dependencies from a teacher model to a student model, enhancing the student's ability to recognize complex label correlations.

To distill label correlations, the paper proposes Batch-level Multi-Order Label Pair Dependency Distillation and Instance-level Multi-Order Label Pair Dependency Distillation to transfer knowledge from both batch and instance, as well as with the consideration of multi-order dependencies (derived from second-order dependencies).

Finally, the method's effectiveness is validated on the Pascal VOC2007, MSCOCO2014, and NUS-WIDE.

**Strengths:**

The paper is well-structured and easily understandable, introducing two loss functions that can capture high-order label correlations from both batch-level and instance-level perspectives.

The experimental results seem impressive, supported by an ablation study that illustrates the efficacy of the two introduced loss functions.

**Limitations:**

Compared to L2D, the novelty seems lacking, as it mainly involves replacing the label-wise distillation and instance-wise distillation loss with batch-level and instance-level dependency loss. More explanation is needed on the advantages of these two proposed loss functions over L2D's Label-wise Embedding Distillation.

The superiority of MDKD in capturing high-order label dependencies is demonstrated with only a single image (Figure 1), which is less convincing. More visualization examples are needed to strengthen this point.

There is a lack of explanation on the calculation method for multi-label probabilities and the activation function. In lines 325-326, the paper states that "the predicted probabilities of all classes should sum to one," suggesting that the activation function is likely to be **softmax**, which differs from L2D. However, $\mathcal{L}_{\mathrm{MLD}}$ is directly referenced from L2D. It is necessary to clarify the activation function and explain the differences in probability calculation compared to L2D.
The phenomenon where the student's performance surpasses the teacher's after distillation (as seen in Table 1) is not reasonably explained. The reviewer would like to see the explanation for such experimental results.

In Figure 3(b), it appears that only the first instance is distilled, but not the b-th instance. In fact, the teacher should distill the b-th instance as well, but the consistency is not depicted, which may lead to misunderstanding.

**Suitability:**

3

---

### Official Review · Reviewer_7J6n · 2024-05-25

**Rating:** 4
**Confidence:** 4

**Summary:**

The manuscript presents a novel framework for Multi-Order Label Pair Dependencies Knowledge Distillation (MDKD) aimed at improving multi-label image classification by explicitly distilling knowledge, on both second-order and high-order label dependencies. This method is tested on several well-known datasets, showing promising improvements over existing methods.

**Strengths:**

1. The proposed MDKD framework introduces an innovative way of considering not only direct label pair dependencies but also higher-order indirect relationships.

2. The authors provide extensive experimental results to validate the effectiveness of the proposed method.

3. The paper is well-written and organized.

**Limitations:**

1. How does the method mine the multi-order co-occurrence? Sequentially averaging the
batch-level second-order label pair dependencies could do this? The authors should give more explanations for that.

2. What's the benefit of the use of Huber loss? No further discussion can be found in this paper.

3. Is it an innovative contribution of this paper for using the Huber loss to measure the second-order label pair dependencies? If not, why not cite relevant literature?

4. The comparison methods seem to be outdated, whether it is possible to add more recent methods?

5. What is the computation complexity and the train/test time of the proposed method?

6. Is the teacher net pre-trained on multi-label datasets, please provide the details. How to get the final prediction results, by the student or fusion of S and T?

**Suitability:**

2

---

### Official Review · Reviewer_j5jM · 2024-05-25

**Rating:** 4
**Confidence:** 4

**Summary:**

This paper introduces the Multi-Order Label Pair Dependencies Knowledge Distillation (MDKD) framework, an innovative approach for multi-label image classification. It aims to address the limitations of existing knowledge distillation methods by explicitly distilling dependencies between labels, including both second-order and high-order dependencies. Therefore, it manages to capture more nuanced label correlations. The paper validates its approach through extensive experiments on datasets such as Pascal VOC2007, MSCOCO2014, and NUS-WIDE, demonstrating superior performances of MDKD.

**Strengths:**

The strength of this paper lies within its focus on an important, yet often-overlooked aspect of multi-label knowledge distillation. While the field of knowledge distillation is well-explored in the context of single-label classification, the area remains significantly under-researched for multi-label settings. The authors have made a commendable effort by exploring and making advancements in this challenging aspect.

**Limitations:**

1. While the experiments are overall sufficient, I find the performance gap between the teacher and student models to be too minimal. This leads me to doubt whether the baseline for the teacher model is set solid enough. It is my strong suggestion to increase the input size from 224 to 448. My recommendation is based on the observation that the COCO dataset tends to yield less stable results with smaller input sizes. I also suggest adding the results from the study 'Knowledge Distillation from Single to Multi Labels: An Empirical Study' to your paper. It provides solid baselines for multi-label KD with a simple method.
2. From Figure 2, the CAM visualization for the L2D method appears to be inferior to that of the vanilla method, which seems counter-intuitive. How would the authors interpret this? Would it be possible to include more visualization figures in the paper for further clarity?
3. How does this approach deal with the issue of missing labels, especially considering that missing labels are a common occurrence in the field of multi-label classification?
4. How would this method perform in scenarios with a larger number of labels? The paper currently only validates the approach with 80 labels; how can the authors ensure the method's generalizability beyond this?
5. I would suggest that, to improve readability and comprehension, the authors could provide an illustrative example and pseudo code of the core method, akin to what was offered in the "RKD" paper. This enhancement can significantly aid readers in understanding the method better.

I am open to further improving my score if the authors can adequately address the above-mentioned concerns and questions.

**Suitability:**

3

---

### Meta-Review · Area_Chair_jJrJ · 2024-06-30

**Recommendation:** Accept (Poster)
**Confidence:** 4

**Metareview:**

- The paper proposes the MDKD framework for the task of multi-label image classification by explicitly distilling both second-order (direct) and high-order (indirect) label pair dependencies from a teacher model to a student model and, therefore, enhancing the student's ability to recognize complex label correlations.

- We have two sufficiently positive reviews and two slightly negative reviews.

- In my opinion, the idea is fine and interesting. It is suggested that the authors carefully revise this paper based on each reviewer's comment in the final version.